# Economic Risk Potential of Infrastructure Failure Considering In-Land Waterways

Rebecca Wehrle [1,*], Marcus Wiens [2], Fabian Neff [1] and Frank Schultmann [1]

1 Karlsruhe Institute of Technology, 76131 Karlsruhe, Germany
2 TU Bergakademie Freiberg, 09599 Freiberg, Germany
* Correspondence: rebecca.wehrle@kit.edu

**Abstract: Purpose**—Unreliable transport infrastructure can cause negative externalities for industries. In this article, we analyze how the private sector is affected by infrastructure failure of public transport infrastructure, using waterways as an example. **Methodology**—To investigate the affectedness of riparian industries, we chose two complementary parallel approaches: A proximity analysis via GIS, and a concluding survey among the identified waterway-dependent industries. An exemplary application is validated by stakeholders. **Findings**—We identify a predominance of location preferences in dependence on waterways for mining, chemical, and metal industries. Their risk tolerance exhibits potentially severe impacts on industries if reliable transport cannot be ensured via waterways, as our paper provides essential insights into the relationship between infrastructure failure and company decisions. Most importantly, we reveal that a lack of alternatives due to missing capacities of other transport modes causes realistic threats to business locations. **Practical implications**—include that a regional focus is crucial for the empirical risk assessment of transport infrastructure. Hence, the data collection should relate to the regional focus groups, particularly the directly affected industries. In addition, infrastructure maintenance should integrate a risk focus and consider the short and long-run impacts on industries.

**Keywords:** waterways; empirical; GIS; location analysis; risk perception

## 1. Introduction

Transport infrastructure as a backbone of modern societies is characterized by interdependencies among infrastructures [1,2], whereas threats of all kinds can lead to systemic and cascading risks [3]. Possible threats are catastrophic events, such as natural disasters or terrorist attacks, but can also arise from human-technical failure, where neglected maintenance may exacerbate critical impacts of catastrophic events.

The example of Inland Waterway Transport (IWT) as a barely studied type of infrastructure demonstrates a potentially dangerous, systemic set of problems in the asset stocks of transport infrastructure, which results in a steadily deteriorating condition of existing transportation infrastructures, as can be observed all over the world [4–7]. A deteriorating transport infrastructure mostly affects the neighboring industries, since cargo must be shifted to other modes of transport, and urgently needed goods experience delivery problems. Depending on the type of goods, different industries can be affected in different ways, such as electricity supply versus chemical products, for example.

As a consequence, deteriorating conditions of construction assets pose a threat to business locations [8]. A variety of options for action comprises operational, organizational, fiscal, and regulatory measures which determine reactions of business decisions [9]. Thus, a systematic assessment of the impacts of infrastructure failure on industries and the broader economy is required, which poses the motivation for our research. Recent investigations identify influencing factors and assess logistics costs, but often neglect the option of relocating whole company locations as reaction to decreasing availability of actually accessible infrastructure (cf. Section 2.3.3).

In this context, we not only examine from a macroeconomic perspective which types of industries favor inland navigation, which is mainly based on freight statistics, but in addition, conduct a microeconomic analysis of location decisions. Hence, we provide an innovative approach of combining macro- and microeconomic analyses with the aim of examining cause-effect relationships with the cause of infrastructure failure and effects as resulting company decisions. Instead of deriving one single key figure, which could be prone to bias and may be misleading, reflecting the complex construct of risk, we derive cause-effect relationships that are critical to understanding the potential for economic risk, which stems from business, and thus human, decisions. We investigate the specific locations affected by disruptions in order to understand and forecast individual economic reactions to infrastructural disruptions. We state the key questions of our research:

1. Which industries and company locations are directly affected by IWT failure?
2. What business decisions may result from lasting availability reductions of IWT?

Figure 1 summarizes our research design: starting from IWT failures, which are characterized by their duration and availability, we proceed to the macroeconomic level and examine freight statistics. We use these to validate the microeconomic GIS-based proximity analysis, enabling the identification of sector-specific location preferences and thus of the affected industrial sites. Subsequently, we use empirical studies to examine the relations between company decisions and failure characteristics. This includes online surveys and interviews. Overall, these studies provide us with the opportunity to identify cause-effect constellations that represent an economic risk potential.

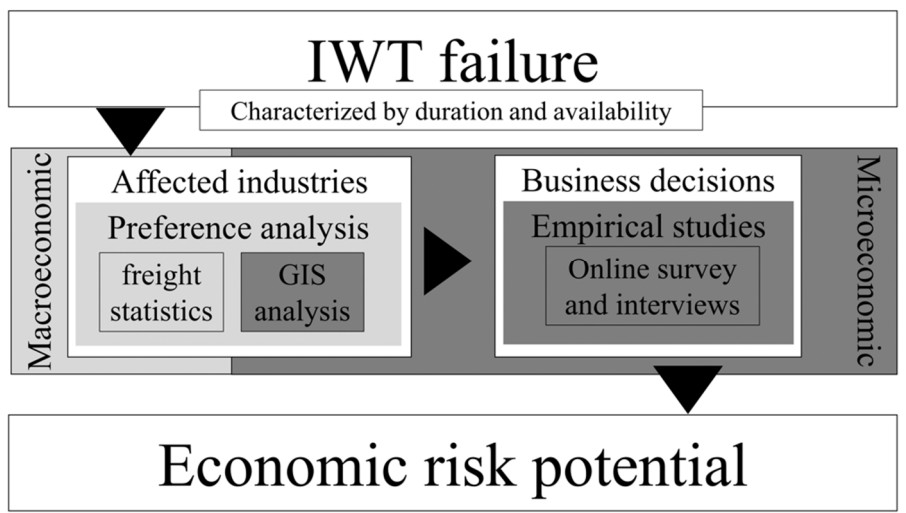

**Figure 1.** Research design.

The paper is structured as follows: We first motivate and explain the underlying problem based on a literature review about the risk assessment of transport infrastructure and the interdependencies toward Supply Chains (SCs) in the following section. Based on this, we derive and apply methods to understand risk perception and decisions of companies resulting from infrastructure damage (Section 3.2). This includes GIS-based proximity analyses, a survey, and the application of expert interviews. The developed methodology is applied to a case study in Germany in Section 4 to highlight the feasibility and relevance of the approach. Finally, we elaborate on the findings and conclude with a critical discussion.

## 2. Literature Review

### 2.1. Externalities of Transport Infrastructure

Externalities arise from economic decisions and activities having an uncompensated impact on uninvolved parties. Negative externalities of transport include infrastructure

damage, accidents, congestion, oil dependence, and environmental externalities such as pollution [10–12]. Further externalities can be observed concerning house prices and rents [13,14], while only a few studies concern waterways and mainly focused on maritime transport [15].

Nevertheless, effects on riparian industries are neglected in the literature, which is why we examine the externalities caused by infrastructure failure as it pertains to commercial transport. Previous studies about the impacts of infrastructure developments on economic productivity are primarily considered only from a growth perspective [16–18], revealing positive externalities of infrastructure investments, such as the productivity of firms and economic agglomeration [19,20]. Ref. [21] provide one of the few studies on water-based transport and calculate savings of travel times caused by improved seaport infrastructure. However, so far, considered external costs of infrastructure use and provision do not comprise externalities of unavailable infrastructure. To this end, negative externalities were analyzed for major road and railway transport. However, to the best of our knowledge, the nexus between the unavailability of waterways and the resulting economic cost for directly affected industries has not been investigated so far.

### 2.2. Risk Assessment of Waterways Infrastructure

#### 2.2.1. Inland Waterway Transport

While IWT is of great importance in nearly every country in the world [8,22], we focus on Germany as an example region. Approximately 18 million tons of goods are transported on German waterways per month [23], while further existing capacity reserves must be used in the future to shift traffic from road to IWT since it is a comparatively environmentally friendly mode of transport [6,22]. IWT thus represents an elementary component of German and European logistics chains that at the same time serve as regional water management in the areas of drinking and service water supply, irrigation, power plant utilization, wastewater disposal, and flood protection for the riparians. Furthermore, waterways fulfill an ecological biotope function and have a high recreational value for people [8].

IWT is a reliable mode of transport when it is in regular operation [4], assuming the full functionality of the infrastructure, and thus of all structures involved, for which predominately government and administration are responsible. However, structures that are system-relevant for the operation of inland navigation are in an increasingly poor condition. They are characterized by a massive maintenance backlog caused by a long-lasting investment deficit [4]. Already in 2015, for example, around 85 percent of locks, 73 percent of weirs, and 87 percent of pumping stations were in an inadequate state of repair [24].

#### 2.2.2. Criticality Assessment of IWT

The literature provides a variety of analytical approaches to risk and related concepts, such as reliability, criticality, and resilience, in the context of infrastructures. A comprehensive review of this is provided by [25]. More recent contributions about the risks of water-related infrastructure are provided by [26], who examine pipe breaks, and [27], who evaluate marine structures, for example. Hence, the number of relevant research objects decreases when specifying the type of infrastructures toward transportation infrastructures, water infrastructures, along with the rare combination of waterway infrastructures, respectively, as Figure 2 illustrates. Moreover, only a few elaborations focus on the economic risk potential of IWT with its potential cascading failures. Below Figure 2, we elaborate on relevant publications in the delimited fields of research.

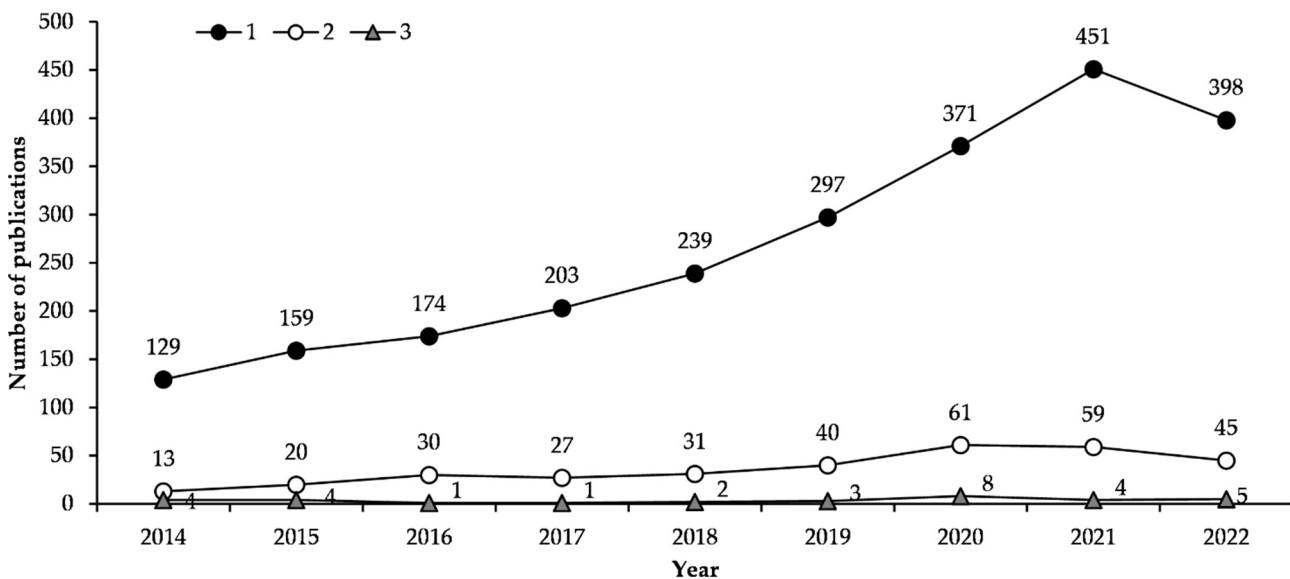

**Figure 2.** Literature on risk-related terms and (1) transport infrastructure, (2) water and transport infrastructure, and (3) waterways infrastructure (Search results from Web of Science, 17 August 2022. 1: ("risk" or "resilience" or "criticality" or "reliability") and "transport*" and "infrastructure" (Topic) and 2014–2022 (Year Published); 2: 1 and "water" (Topic); 3: 1 and ("waterway*" or "IWT") (Topic)).

IWT, as part of the critical infrastructures (CIs), provides fundamental services that are substantial to the safety, as well as the economic and social welfare, of a society [2]. While vulnerability refers to the hazard-specific susceptibility of a system to impairment or failure of its functionality [28], the resulting criticality refers exclusively to the consequences of a system failure, independent of the probability of occurrence [29].

Harmful consequences of impaired infrastructure as negative externalities refer to various dimensions of criticality, such as the economic, structural, social, as well as ecological dimensions [30]. In this paper, we address the economic dimension of criticality. Ref. [31] focus more on societal impacts than on economic consequences, while their scenario-based risk quantification uses a psychometric 4-point Likert scale, which is based on various literature reviews, expert opinions, and statistics. Ref. [32] use the parameters of frequency, probability, extent, and duration to assess the risk of CI, assuming an infrastructure failure as an initializing scenario leading to a subsequent loss of one or more societal critical function(s). Ref. [33] assess the criticality of highway transportation networks, considering congestion effects caused by the interaction of traveler behavior and the built environment [33]. Ref. [34], moreover, consider the interactions of different CI sectors.

There are just a few approaches which come close to our research objective: the assessment of economic risk potential resulting from infrastructure failure of IWT. While the literature recognizes the harmful potential of water-related infrastructure assets, we find a focus on dam breaks (e.g., [35,36]). We also observe literature on economic losses due to flooding, but which examine specific cases like tsunamis [37–39], which have nothing to do with artificial waterways and mostly affect coastal regions, and whose vulnerabilities and exposure differ from inland structures [40].

Investigations about IWT include those from [41,42] who provide stochastic measures of infrastructure network resilience but neglect the critical outcome of network disruptions. Ref. [43] assess the robustness of multimodal freight networks by network measures. Ref. [44] examine the resilience of the transport system under consideration of ship load, ship delay, and recovery cost. Hence, they focus on recovery as significant resilience capacity [5].

Ref. [45] address the system of systems character of IWT within a risk assessment framework based on a multi-level approach, which allows the integration of both structural

vulnerability and empirical research as economic risk potential. This understanding of risk forms the basis for the understanding of risk used in our research. Conversely, our research is predominantly a micro-level approach, which examines and integrates corporate decisions in order to be able to derive macro-perspective conclusions.

*2.3. Supply Chain Management and Dependency on Transport Infrastructure*

To assess the economic risk potential resulting from infrastructure failure, it is vital first to understand industrial vulnerability and criticality in order to identify affected industries and their reactions toward disruptive events. Therefore, we elaborate on SC Risk Management (SCRM) and the impacts of disrupted transport infrastructure on SCs before we come to proximity analyses to determine the concerned industrial sectors.

2.3.1. Supply Chain Risk Management and Risks as Disruptive Events

An SC is a system consisting of several companies that are directly or indirectly involved in the fulfillment of customer needs and thus includes not only manufacturers and suppliers, but also transport service providers, wholesalers, and retailers, as well as customers. The linkage of the enterprises comes about through flow orientation, i.e., by the flow of information, money, and goods [46]. SCRM, as a specific domain of SC Management, deals with risk identification, the assessment of possible damages and reduction, as well as mitigation of the impacts of threats to the SC by using control measures [47,48].

SC risks relate to a possible discrepancy between supply and demand and the resulting consequences and may arise from three different risk sources: (1) environment-related risks, which arise due to the interaction of the supply chain with the environment (e.g., floods, terrorist attacks), (2) organizational risks within the boundary of the chain (e.g., strikes or machine failure), and (3) network-related risks that occur due to interactions of firms within the supply chain [49]. Since SCRM aims to enhance and ensure resilient SCs, possible disruptive events must be handled as risks, whereas infrastructure disruptions differentiate network-related risks, according to [49].

Furthermore, SC risks include procurement and sales risks, among others, while we set the focus on transport risks and the interdependencies between risks. Besides unavailable infrastructure, transport risks primarily affect procurement and sales since freight cannot be transported in due time. Regarding these interconnected risks, it must be noted that infrastructure is the essential factor in the logistics performance of companies [50]. Thus, measures to enhance logistics performance and reduce threats to this performance prerequisite that infrastructure takes a significant role within the scope of companies' SCRM.

2.3.2. Proximity of Business Locations and Transport Infrastructure

In order to determine the industrial risks of failing transportation infrastructures, it is initially important to analyze which companies are dependent on them, leading to sub-question *(1) Which industries and company locations are directly affected by IWT failure?* This can be done, on the one hand, by using freight transport statistics, which do not allow any conclusions to be drawn about specific companies. Therefore, it is necessary to have a closer look at specific locations and their accessibility to various infrastructures.

Location decisions are long-term investments and thus of great importance in the strategic planning of companies [51], since transport links have an essential role in the location decision of industrial companies [52]. Depending on the industry, industrial companies prefer different means of transport for freight traffic [53]. While the topics of company location decisions and freight transportation mode choice are widely studied in research, their interrelationship, in contrast, has been barely studied, as illustrated by Table 1. A closer look reveals that most of the literature on distance determination of infrastructures to companies refers to passenger transport rather than freight transport [54–56]. Ref. [57] consider only freight transport, but do not calculate the distance to the access points of the transport systems, instead assessing the accessibility of Belgian regions based on distances in the road, rail and waterway network.

**Table 1.** Literature concerning business locations and proximity to transport infrastructure [1].

| Literature | Location Planning | Transportation Choice | Spatial Structure | Company | Branch | Access Points | Distance Measurement | Goods |
|---|---|---|---|---|---|---|---|---|
| [58] | | X | | X | | | | X |
| [59] | | X | | X | | | | X |
| [60] | | X | | X | | | | X |
| [53] | | X | | X | X | | | X |
| [61] | | X | | X | | | | X |
| [62] | | X | | X | X * | | | X |
| [63] | | X | | X | X * | | | X |
| [64] | | X | | X | X * | | | X |
| [65] | | | X | X | | | | |
| [66] | | | X | X | | | X | |
| [67] | | | X | X | | | X | |
| [68] | | | | | | X | X | |
| [69] | | | | | | X | X | |
| [70] | X | | X | X | X | X | | X |
| [71] | X | | | X | | X | | |
| [72] | X | X | X | X | | X | | X |
| [73] | X | | X | X | X | X | X | |
| [54] | X | | X | X | X | X | X | |
| [55] | X | | X | X | X | X | X | |
| [56] | X | | X | X | X | X | X | |
| [74] | X | | X | X | X | X | X | |
| [57] | | | X | | | X ** | X | X |

Notes: [1] * Differentiation of goods segments instead of industries; ** Consideration of transport networks instead of access points.

### 2.3.3. Impact of Transport Disruptions on Business Activities

After the identification of sectors concerned, we want to understand which kind of business decisions may result from infrastructure failure of IWT. This objective is summarized by sub-question *(2) What business decisions may result from* lasting availability reductions of IWT?

Therefore, the procedures, occurrences, and impacts of IWT disturbances must be examined. Existing evidence reveals that large disturbances of waterways occur again and again, such as blockades of the Suez Canal [75], while 12% of the world's total ocean trade traverses the Suez Canal [76]. Among others, Sony faced a combination of miscalculation and delivery delay, which caused a 90% decrease in the PlayStation 2 sales volume in Great Britain [77], and a blockade in 2021 revealed blocking costs of $400 million per hour [78].

Impacts of disrupted SCs can be classified into (1) direct, i.e., physical damages, and (2) indirect effects, which include all ripple effects. Whereas direct effects in a water-related context rather arise from floods since the effects include destroyed inventory or machines, infrastructure failure is assigned to the indirect effects [37].

Measures of SCRM involve changes in the SC structure and the choice of transportation routes and means [79]. In moving toward a more resilient SC, SC visibility should be increased, buffer capacities could be enhanced, and companies could consider reorganizing the structure of the SC, such as going from single-sourcing to multi-sourcing or considering the use of more generic input components [80–82]. While alternative routing may be

sufficient for short-term disruptions, there is, in most cases, no feasible solution in the long term, such as shifting IWT cargo toward other means of transport, which are regularly capacity-restricted.

For specific goods, it is possible to become independent from public infrastructure via own pipelines [83]. However, for most industries and goods, this is not a feasible option. As a matter of fact, maintaining highly infrastructure-dependent facilities can be more costly than relocating them to another site [84]. This can especially come into focus when companies are confronted with degrading infrastructure, since reliable transport infrastructure plays a vital role in companies' location decisions and is the most important factor in logistics performance [50,74].

Literature revolving around this problem and the scoping of corporate decisions can be found in the field of SC management and more specifically in the fields of risk and crisis management dealing with SC disruptions [85]. The latter can be caused by pandemics, terrorist attacks, labor strikes, and man-made defections, for example, and can cause production or transportation stops [86]. Literature about strategic implications, such as location planning, is relatively scarce [87–89], and existing research in this area often neglects the context of short-term disruptions since supplier network design is mostly emphasized [90,91]. Moreover, external influencing factors find little attention.

By extending the literature research toward Facility Location Planning (FLP) and transport infrastructure, we encounter a focus on the spatial structure of companies and infrastructure, as well as on location(s) for new site(s), rather than on the relocation of existing facilities [70–73]. Moreover, literature about the relocation of facilities arose but it is still scarce with regard to infrastructural aspects, focusing more on accessibility as the predominant infrastructure factor in relocation decisions [54,92].

As demonstrated before, the literature on risk, related terms, and IWT is already scarce (cf. Figure 2 and subsequent elaborations). Hence, investigations on interdependencies towards related SCs and business reactions are similarly rare. Among the notable exceptions we find [93], who analyze the effects of IW port disruptions on SC networks and vice versa by using Bayesian networks. Ref. [9] identify cost, time, reliability, flexibility, and environment as the most significant factors for barriers to intermodal freight diversion, and assess total logistics costs of dominating transportation modes. The investigations of [94] provide an impetus for the evaluation of economic effects resulting from infrastructure failure since they estimate economic losses stemming from dam/lock failures, but only consider coal delivery.

## 3. Research Methodology

### 3.1. Concept

Our methodological approach determines the economic risk potential of unavailable IWT. Figure 1 already introduced the concept of our approach. In particular, we analyze the potential impacts of infrastructure failure on the economy, using a GIS-based proximity analysis and empirical studies, namely a survey supplemented by expert interviews, to achieve background information about corporate decisions and risk perception.

### 3.2. Economic Risk Potential

To assess economic risks resulting from infrastructure failure, it is first essential to assess the general hazard potential of the infrastructure in the (supra-)regional area with regard to the industry. We suggest a combined approach of a proximity analysis of facilities toward infrastructure access points (APs) to identify dependent industry facilities, and empirical studies to elaborate on the data and background of self-reported industrial vulnerability and criticality. Overall, our research concept allows us to anticipate economic stakeholders' reactions toward infrastructure failure. This, in turn, reflects our aspired economic risk potential as insights into cause-effect relationships.

### 3.2.1. Proximity Analysis

According to our first research question (*Which companies are directly affected?*), we analyze which business facilities rely on IWT by an analysis of spatial proximity based on the relationship between location decisions, spatial structure, and modal choice of freight transport across all modes of transport. Location preferences of industrial facilities are determined depending on the affiliation to industrial sectors and are based on the central assumption that higher preferences correlate negatively with the distance between industrial sites and transport system APs. To assess spatial proximity, the distance between industrial sites and transport system APs is calculated using geographic information systems (GIS). Figure 3 illustrates the process, which is explained in more detail in the following.

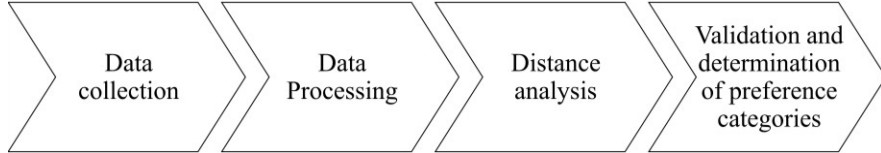

**Figure 3.** Process of proximity analysis.

Firstly, infrastructure network data and location coordinates of the APs of further transport systems, as well as data about industrial sites, are collected, containing locations and industry classifications in the area of interest. In Step 2, the edges of the infrastructure network are assigned travel times as edge weights, calculated by route lengths and assumptions regarding the average speed of infrastructure categories.

The distance analysis (Step 3) is performed using Dijkstra's algorithm to calculate the shortest travel times for given start nodes (industrial sites) and destination nodes (infrastructure APs: highway interchanges, railroad terminals, public ports, and airports; radius consideration of company-owned railway and ports). For each industry, the mode-specific average travel time $t_m$ from companies to APs of transport mode $m$ is calculated and taken as a measure of spatial proximity, of which the preferences of the industries for the transport systems are derived in Step 4: by identifying correlations between the industry-specific averages of travel times and the transport volumes from freight statistics, conclusions can be drawn whether shorter travel times actually correspond to a higher preference in use. For each transport system, preference categories are derived, using the relative deviation from the mean value of the travel times of all industries:

$$IB_{i,j,m} = \left(1 + d_{i,j}\right) \times \bar{t}_m \tag{1}$$

The interval boundary $IB_{i,j,m}$ indicates the value of the travel time that delimits category $i$ from category $j$, using $t_m$. Thus, negative values for the relative deviations $d_{i,}$ mean shorter travel times compared to the average and thus correspond to a higher preference. If a validation with freight goods statistics reveals many inconsistencies in the achieved classification, no valid preferences can be derived for the transport system under consideration. Otherwise, it is permissible to derive preferences, as shown by the classification of the preference categories.

Using this procedure, it is possible to identify sectors with exceptionally high sector preferences which (1) are strongly dependent on specific infrastructure and their APs, and (2) indicate corresponding location preferences in the survey. Accordingly, the sectors with a preference for IWT are most affected by interruptions and need special attention in understanding the influence of disruptions on business decisions regarding IWT disruptions.

### 3.2.2. Empirical Studies

After reflecting location preferences of companies on a macroscopic level to identify concerned industries, we address sub-question *(2) What business decisions may result from lasting availability reductions of IWT?*

Since business decisions like modal shifts or (re-)locations of facilities have significant impacts on business locations, it is essential to understand the interconnections between infrastructure disruptions and business decisions. Therefore, we propose to proceed as illustrated in Figure 4. Firstly, formulate key questions and derive hypotheses. In the second step, we address these using an (online) questionnaire containing quantitatively and qualitatively evaluable questions to gather further data directly from decision makers of companies. In the context of industrial criticality, we examine the following two key questions: (1) How do companies perceive a decrease in availability of IWT, and (2) How does this affect their business activities?

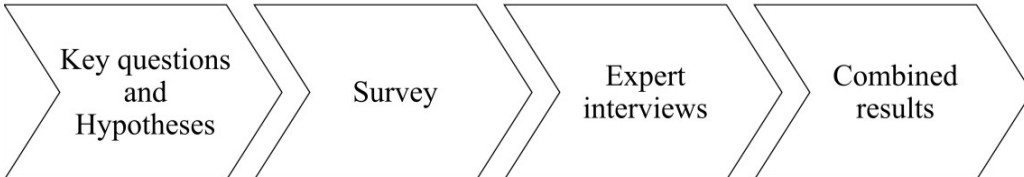

**Figure 4.** Process of empirical studies.

Beyond the survey, more in-depth knowledge about motives and experiences is gathered through targeted expert interviews with survey participants, addressing the formulated hypotheses while maintaining the reference to the questionnaire (Step 3). In the last step (4), the results of the survey and interviews are combined to provide deeper insights into cause-effect relationships.

## 4. Economic Risk Potential of Infrastructure Failure in Case of the West German Canal Network

### 4.1. West German Canal Network

We apply our methodology to the area of the West German Canal Network, which consists of 350 km of canals connecting the Ruhr area and the German North Sea ports [95]. In 2013, the volume of goods transported on the West German canal network amounted to approximately 226.8 million freight tons, which corresponds to 37.1% of the total waterborne transport volume in Germany [96]. Moreover, the region of North Rhine-Westphalia (NRW) is characterized by a comparatively high population density and significant economic importance, which is mainly due to two sectors: coal mines and the steel industry.

### 4.2. Economic Risk Potential

#### 4.2.1. Proximity Analysis

**Steps 1 and 2: Data Collection and Processing**

The selected modeling region causes a limitation for the analysis of companies in NRW, with [97] being the primary source for data collection. Data about industrial sites is filtered manually and supplemented with information on the industrial classification from Table 2 [98]. Table 2, moreover, shows the numbers of company locations considered with a total of 2823, which are illustrated in Figure 5.

**Table 2.** Industrial categorization. * [98].

| Category | Name | Section * | Division * | Number |
|---|---|---|---|---|
| B | Mining and quarrying | B | all | 44 |
| C1 | Production of food and feed, beverage production | C | 10, 11 | 137 |
| C2 | Coking plant and mineral oil processing | C | 19 | 15 |
| C3 | Production of chemical and pharmaceutical products | C | 20, 21 | 61 |
| C4 | Manufacture of rubber and plastic products | C | 22 | 80 |
| C5 | Manufacture of glassware, ceramics, processing of stones and earths | C | 23 | 118 |
| C6 | Metal production and processing | C | 24 | 59 |
| C7 | Production of metal products | C | 25 | 325 |
| C8 | Manufacture of computer, electronic and optical products, manufacture of electrical equipment | C | 26, 27 | 106 |
| C9 | Mechanical Engineering | C | 28 | 238 |
| C10 | Manufacture of motor vehicles and parts of motor vehicles | C | 29 | 45 |
| CX | Other manufacturing | C | 12–18, 30–33 | 221 |
| D | Energy supply | D except DX | all | 30 |
| DX | Biogas and solar plants | D | additionally defined | 23 |
| E | Water supply; wastewater and solid waste management & pollution cleanup | E | all | 453 |

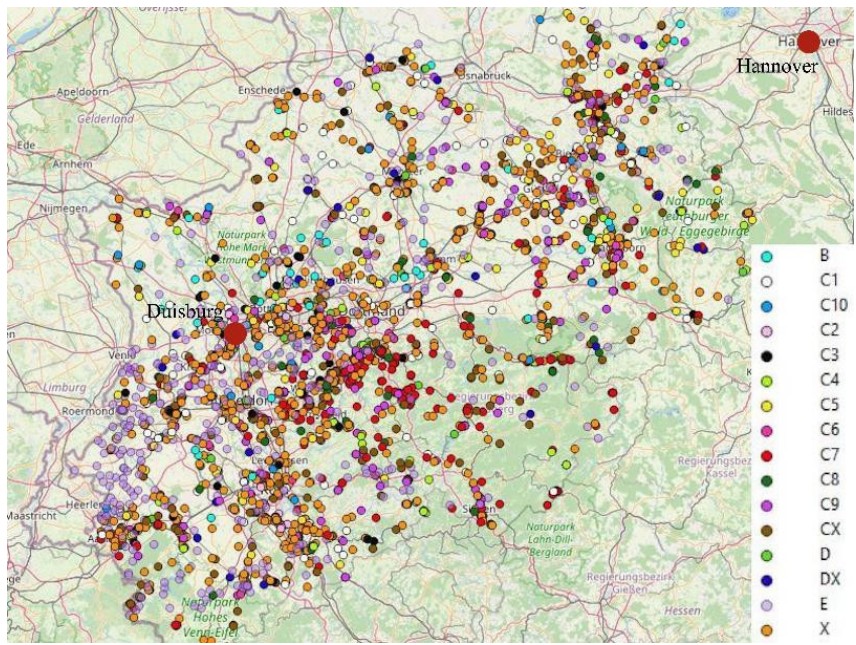

**Figure 5.** Company locations by industry category.

Peripheral infrastructure was also captured with road network data extracted from [99] as Figures 6 and 7 visualize, extended by data sources for railroad terminals [100]. The track connections and railroad terminals are depicted in Figure 8. Analogously, the data on ports was extracted, as shown in Figure 9, whereas airport locations were added manually. For better orientation and comparability of Figures 5–9, the cities of Duisburg and Hannover were drawn on the maps.

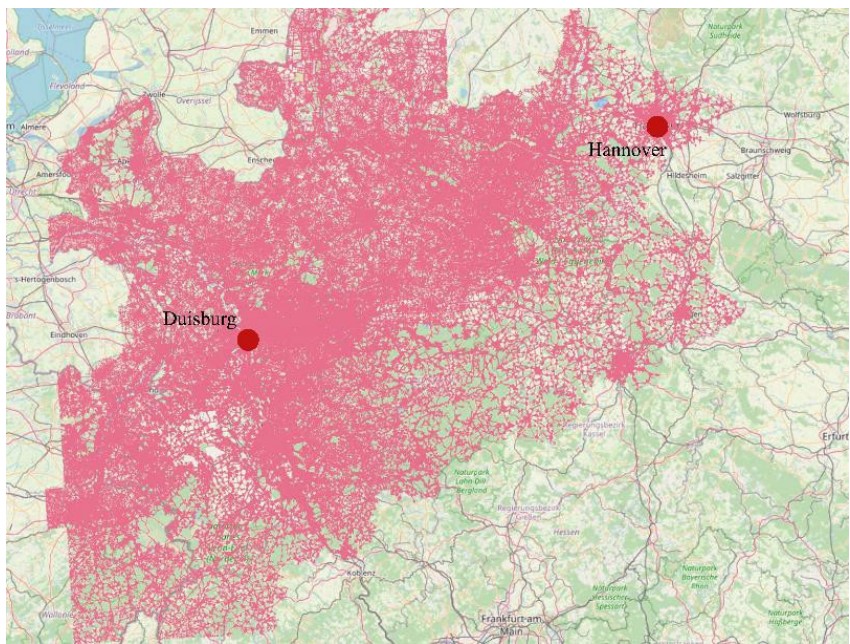

**Figure 6.** Road network.

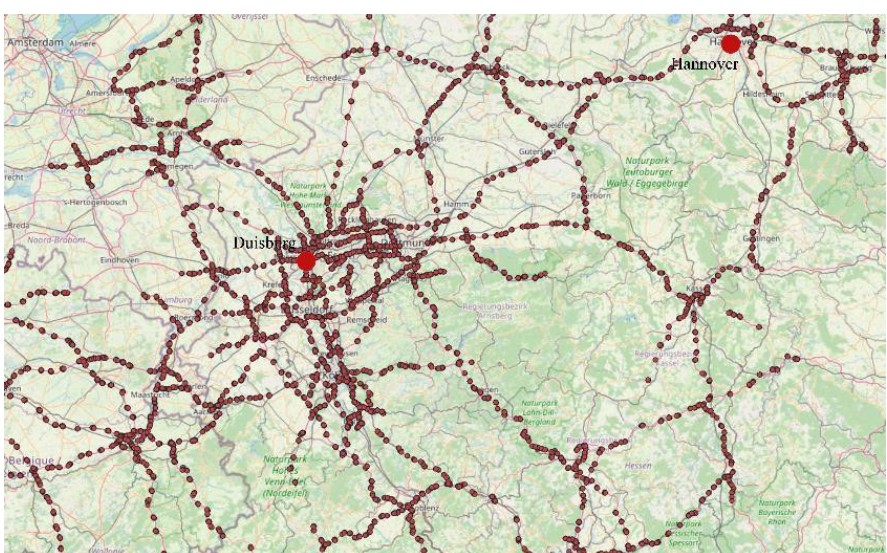

**Figure 7.** Locations of the highway junctions.

**Step 3: Data analysis**

The results of the application of Dijkstra's algorithm and an overview of the results of the radius analysis are given in Table 3, which shows for each industry category (1) the number of companies, and (2) the percentage of companies that were assigned an AP.

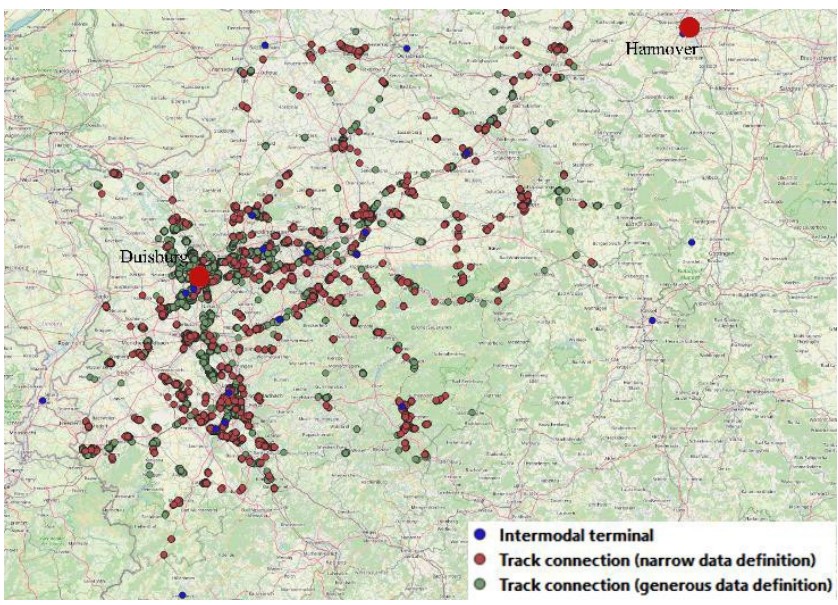

**Figure 8.** Locations of the rail infrastructure.

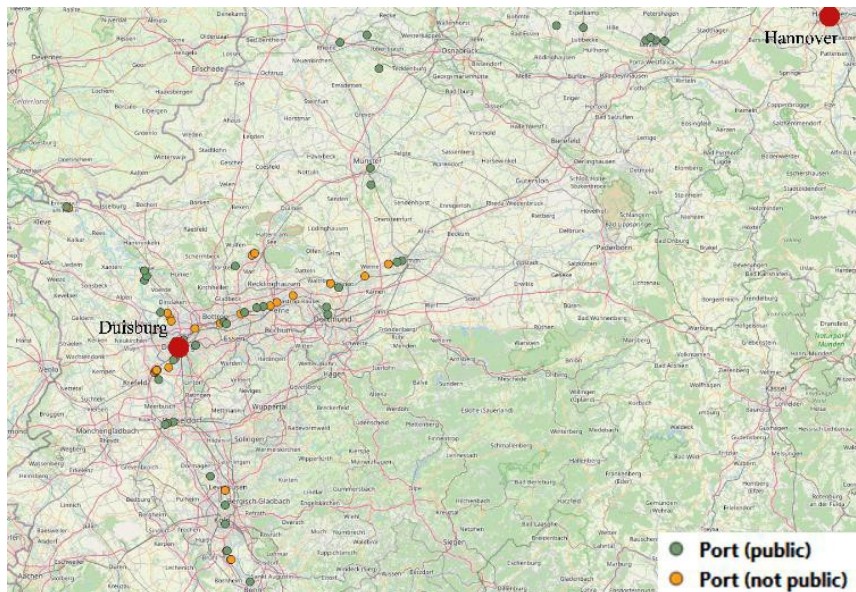

**Figure 9.** Port locations.

The radius analysis is performed with radii of 250 m and 500 m for track connections and 2000 m for ports. The distance to the nearest AP is calculated for each industrial site, whereas an allocation is anticipated if this distance is less than the defined radius. Thus, for each industry, it is possible to determine the proportion of companies that have been assigned a track connection or a port. This result is taken as a measure of spatial proximity. Only the nearest APs are considered for each industrial site to optimize runtime. Then, only the APs with the respective shortest travel times to each transport system are considered. Furthermore, due to ambiguities in the extraction of rail APs from OSM, we use two different databases that include a more generous and narrower definition of track connections, respectively, as Table 3 shows.

**Table 3.** Averages of travel times (Dijkstra) and share of allocations (Radius analysis) by industry category. * different numbers of industry E and X [2].

| Industry Category | Dijkstra | | | | | Radius Analysis | | |
| | Number | Highway | Airport | Railroad Terminal | Public Port | Port (Not Public) * | Track Connection (Generous) * | Track Connection (Narrow) * |
|---|---|---|---|---|---|---|---|---|
| B | 44 | 9.17 min | 45.37 min | 25.27 min | 22.20 min | 4.55% | 29.55% | 15.91% |
| C1 | 137 | 8.77 min | 42.29 min | 29.38 min | 30.68 min | 0.73% | 16.06% | 6.57% |
| C2 | 15 | 6.52 min | 35.58 min | 13.03 min | 8.95 min | 53.33% | 66.67% | 33.33% |
| C3 | 61 | 7.15 min | 41.05 min | 24.94 min | 22.57 min | 13.11% | 54.10% | 29.51% |
| C4 | 80 | 12.45 min | 51.23 min | 33.40 min | 40.86 min | 1.25% | 18.75% | 7.50% |
| C5 | 118 | 10.26 min | 40.89 min | 32.92 min | 36.32 min | 2.54% | 22.03% | 13.56% |
| C6 | 59 | 7.19 min | 42.18 min | 22.72 min | 26.32 min | 8.47% | 50.85% | 20.34% |
| C7 | 325 | 9.36 min | 52.35 min | 30.48 min | 39.86 min | 1.54% | 19.08% | 11.69% |
| C8 | 106 | 10.06 min | 48.95 min | 31.76 min | 34.53 min | 0.94% | 19.81% | 14.15% |
| C9 | 238 | 8.57 min | 44.70 min | 28.04 min | 33.70 min | 0.84% | 23.11% | 9.66% |
| C10 | 45 | 8.02 min | 40.21 min | 30.12 min | 36.02 min | 0.00% | 31.11% | 22.22% |
| CX | 221 | 10.21 min | 47.21 min | 33.37 min | 36.56 min | 0.90% | 15.84% | 10.41% |
| D | 30 | 11.06 min | 50.06 min | 29.26 min | 23.76 min | 13.33% | 56.67% | 46.67% |
| DX | 23 | 13.13 min | 46.36 min | 36.18 min | 34.55 min | 0.00% | 8.70% | 4.35% |
| E | 423 (453) * | 9.56 min | 43.94 min | 31.17 min | 32.58 min | 2.65% | 11.92% | 6.18% |
| X | 898 (912) * | 8.20 min | 43.11 min | 27.85 min | 28.61 min | 3.95% | 28.84% | 15.24% |
| All industries | 2.823 (2.867) * | 9.06 min | 45.06 min | 29.52 min | 32.28 min | 3.14% | 23.44% | 12.69% |

Notes: 2 The OSM road data was used as LineString data and converted into a network with nodes and edges. Individual nodes could not be automatically connected to the network; hence, no route calculation was possible for individual companies. Due to the large sample, no manual post-recording was done for categories E and X, while companies in other categories with smaller data volumes were assigned manually. For the radius analysis, all companies could be considered since its independence of road network data.

**Step 4: Results and interpretation**

The following values for the relative deviations (Negative $d_i$, mean shorter travel times and higher preferences. The choice of values is based on observed distribu-tions and sensitivity analyses of travel time deviations). $d_i$, are proposed in order to achieve five delimitable preference categories by applying Equation (1): $d_{1,2} = -0.15$, $d_{2,3} = -0.05$, $d_{3,4} = 0.05$, $d_{4,5} = 0.15$. In order to obtain a comprehensive picture and to identify contradictions, an analysis of all transport modes is performed. The results generally reveal short travel times to highway interchanges with a non-significant Pearson correlation coefficient between travel times and transport performance from freight statistics (r = 0.280; $p$ = 0.433). The preferences derived from the categorization hardly correspond to the importance of road transport for the respective industries, which can be seen in freight statistics. Because of these inconsistencies, it is not possible to derive preferences for highway infrastructure from travel times.

Travel times to the nearest airport reveal a wide range of data and no industry with particularly short travel times. Since the freight statistics do not contain any data on air freight traffic, data on the export value of goods [101] is used for validation. Nevertheless, a derivation of preferences is not possible, and no classification into preference categories follows.

Figure 10 shows the travel times to the railroad terminals. The coking plant and mineral oil processing industry (C2) stands out with particularly short travel times. The correlation with freight statistics is negative at r = −0.596 (*p* = 0.053), but is not statistically significant at the 0.05 level. Industries are then divided into preference categories (Table 4).

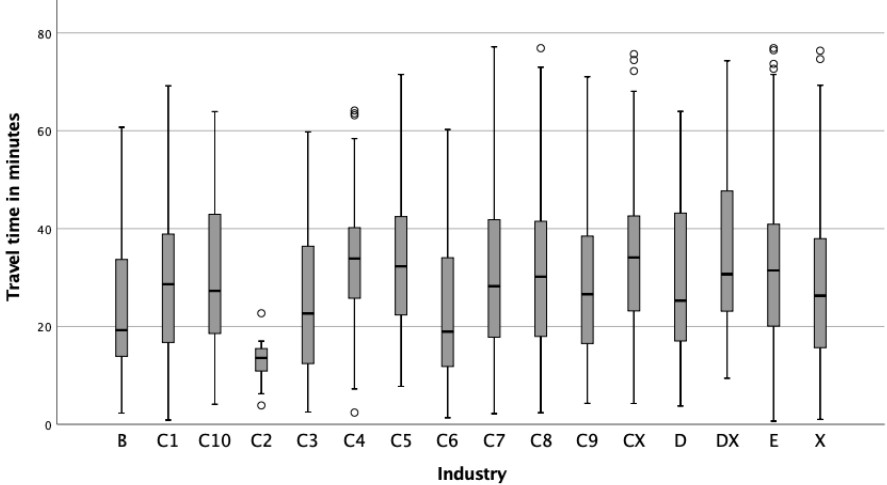

**Figure 10.** Travel time to railroad terminals by industry category. Circles indicate outliners.

**Table 4.** Preference categories and categorization of the industries.

| Preference Category | Assumed Preference | Interval of the Industry Mean Values of the Travel Times | Industry Category | Interval of the Industry Mean Values of the Travel Times | Industry Category |
|---|---|---|---|---|---|
| | | **Railroad Terminals** | | **Public Ports** | |
| 1 | Very large | (13.03 min; 25.09 min) | C2, C3, C6 | (8.95 min; 27.44 min) | B, C2, C3, C6, D |
| 2 | Large | (25.09 min; 28.04 min) | B, X | (27.44 min; 30.67 min) | X |
| 3 | Average | (28.04 min; 31.00 min) | C1, C7, C9, C10, D | (30.67 min; 33.89 min) | C1, C9, E |
| 4 | Low | (31.00 min; 33.95 min) | C4, C5, C8, CX, E | (33.89 min; 37.12 min) | C5, C8, C10, CX, DX |
| 5 | Very low | (33.95 min; 36.18 min) | DX | (37.12 min; 40.86 min) | C4, C7 |

The fact that proximity to the rail infrastructure implies a higher preference is also observed when looking at the companies' own track connections: The share of allocation is shown in dependence on the selected radius and the data basis in Figure 11. Sectors with exceptionally high allocation shares also show significantly higher transport performance in the freight statistics.

Figure 12 shows travel times to public ports. The correlation with freight statistics is strongly negative with r = −0.862 (*p* = 0.001). An overview of the correlations is given in Table 5. Industries are also classified into preference categories for ports and identified preferences can be fully validated by the freight statistics (Table 4). An examination of the allocation shares of non-public ports confirms this classification. The industries in the highest preference category also have the highest allocation shares and high transport performance in the freight statistics.

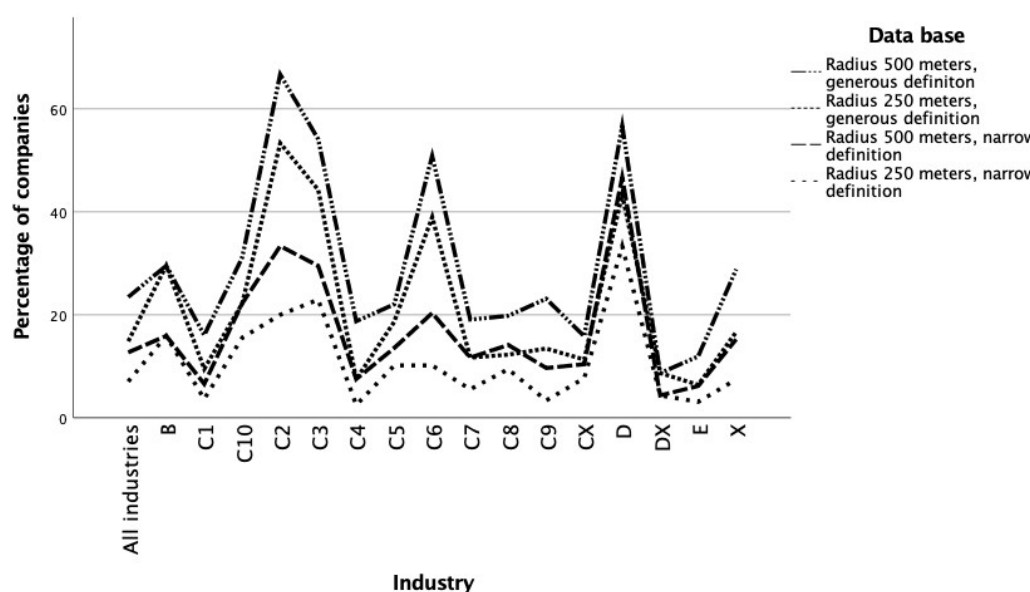

**Figure 11.** Track connections: comparison of different parameters of radius analysis.

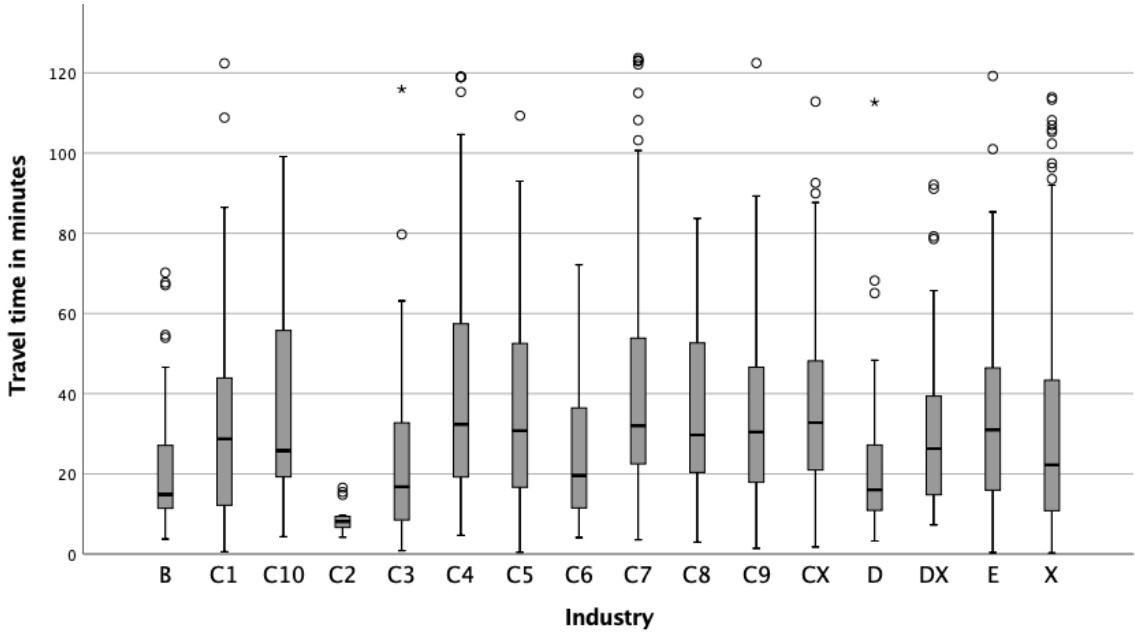

**Figure 12.** Travel times to public ports by industry category. Circles indicate outliners, an asterisk illustrates extreme outliners.

**Table 5.** Correlation coefficients of different transport systems.

| | Parameters | Pearson Correlation Coefficient |
|---|---|---|
| **Freight statistics** | Highway junctions | $r = 0.280$ ($p = 0.433$) |
| | Railroad terminals | $r = -0.596$ ($p = 0.053$) |
| | Public ports | $r = -0.862$ ($p = 0.001$) |

In summary, the proximity analysis only identifies preferences for rail freight and freight shipping. The industries of mining and quarrying, coking plant and mineral oil processing (C2), production of chemical and pharmaceutical products (C3), metal production and processing (C6), and energy supply (D) show high preferences for rail

freight transport and freight shipping. All other industries considered have only average or below average preferences for these two types of transport systems.

Thus, our sub-question *(1) Which industries and company locations are directly affected by IWT failure?* can be answered on the basis of these location preferences. The affected industries and companies have the alternatives of relocating either transports or entire sites if the function of IWT is restricted, which depends on the actual threat, the risk perception, and the feasibility, economic efficiency, and capacities of the alternatives.

### 4.2.2. Empirical Studies

The previous analysis supports the unique role of IWT as a mode of transport and allows us to identify sectors and facility locations that are strongly dependent on IWT. Moreover, location preferences are derived. As these do not only depend on the accessibility to the infrastructure, but also on the availability, and the resulting risk perception and business activities play an essential role, a deeper analysis of these factors and interdependencies is conducted based on the elaborations in Section 3.2.2 and the results of the previous section.

**Step 1: Key question and Hypotheses**

The two key questions are formulated in Section 3.2.2, leading to eight significant research topics which are depicted in Table 6 and serve as the basis for the hypotheses, two of which are shown in Table 7.

**Table 6.** Research topics.

| Nr | Research Topic |
|----|----------------|
| R1 | Flow of goods and supply relationships |
| R2 | Temporal disturbance progressions |
| R3 | Vulnerabilities of various industries |
| R4 | Application of risk reduction measures |
| R5 | Assessment of damage caused by interrupted supply chains |
| R6 | Identification of highly critical event scenarios |
| R7 | Effect of water contamination and shortage of cooling water |
| R8 | Connections with other CI: power supply and water supply |

**Table 7.** Exemplary hypotheses.

| Nr | Research Topic | Hypothesis |
|----|----------------|------------|
| H1 | R1, R4 | For transports currently transported via waterways, there are hardly any alternative options |
| H2 | R2, R5 | It is feared that dependence on IWT will lead to considerable problems in the future |

**Step 2: Survey**

In the first instance, an online questionnaire consisting of qualitative and quantitative, primarily closed, questions was conducted to address the hypotheses. Thereby, the pattern of industries to which the recipients belonged to was quite heterogeneous except the commonality that the represented firms are dependent on IWT. Primarily, entrepreneurs located in the area of NRW were contacted. Thus, we contacted a total of 231 companies and associations with a response rate of only 21 usable responses (response rate 0.09%) (25 responses, removal of 4 due to duplicate records or non-relevance for IWT). To assess the vulnerability of industrial facilities, we asked company representatives about the measures companies are already taking to reduce risks from waterway dependency. In this context, more than 60% rely on a shift of transport mode, as Figure 13 shows. Measures of expanding storage capacity, as well as the use of a redundant water supply, are pointed out by a much lower percentage of companies. The latter is due to the fact that water extraction from the canal network is not carried out by many companies and is not further relevant to transport (H1).

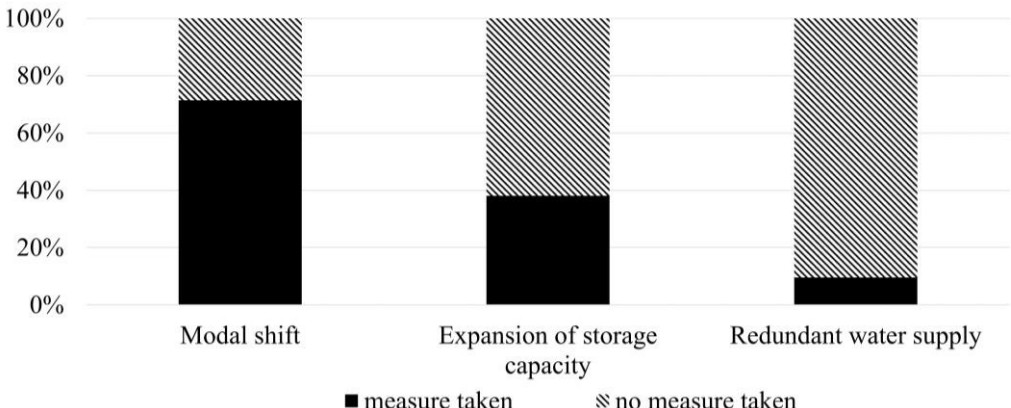

**Figure 13.** Measures to reduce risks arising from waterways.

Moreover, if the measure of modal shift is considered, the following can be observed: Participants that have implemented modal shift measures in the past are more likely to have planned to do so in the future. According to an adjusted *t*-test (Welch test), the likelihood of shifting traffic in the future was found to be 80% higher among those who are currently already doing so than among those who are not currently implementing any modal shift measures.

A closer look at modal shift reveals that, according to the companies' assessment, the short-term shift to road has the largest potential for modal shift, with a medium rating of 40%. Rail plays only a minor role and is hardly available in the short term, as depicted in Figure 14.

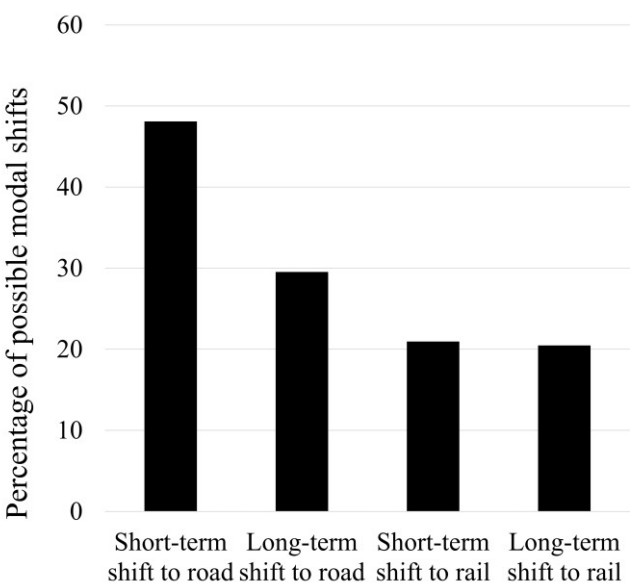

**Figure 14.** Extent of possible modal shift.

The barriers to modal shift (Figure 15) furthermore reveal that the capacity of alternative transport systems, as well as increased costs, are significant factors that stand in the way of modal shift. This is equally true for rail and road, with barriers to modal shift to rail considered to be higher. Consequently, transport that currently takes place via waterways is indeed challenging to shift, especially since the focus of the surveyed firms lies mostly on alternative short-term options, and the barriers to a long-term shift in transport were perceived as too high.

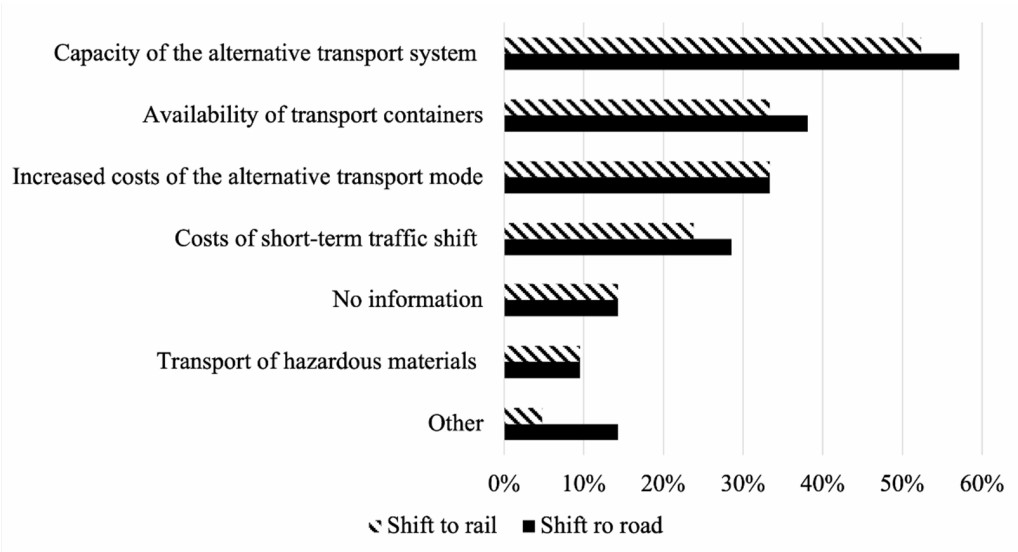

**Figure 15.** Barriers to modal shift.

Moreover, company representatives were asked to rate how much of an impact a 3-week shutdown would have on business operations. The results, illustrated in Figure 16, show that the longer the lead time, the less constrained the companies' ability to do business. With a warning period of one month, no severe problem is initially feared. In contrast, a warning period of only one day leads to a much higher impairment of business activity.

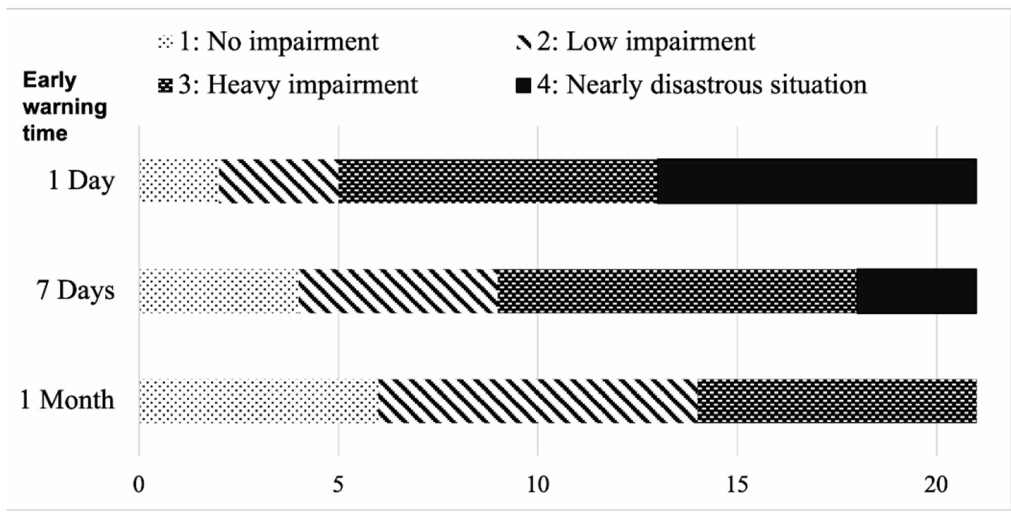

**Figure 16.** Criticality depending on early warning time—closure of three weeks.

An examination of the correlation (Figure 17) with revenue also shows that the higher the company's revenue, the higher the level of constraint as the lead time becomes shorter. Reasons for this could be the more complex infrastructures and processes, and higher flows of goods that larger companies have to convert. They can often only act much less flexibly than smaller companies.

Thus, regarding our hypotheses, H1 could be confirmed. In contrast, H2 could not be confirmed because the risk of the current and future maintenance of the canal system was rated mostly the same over the considered companies, as Figure 18 shows.

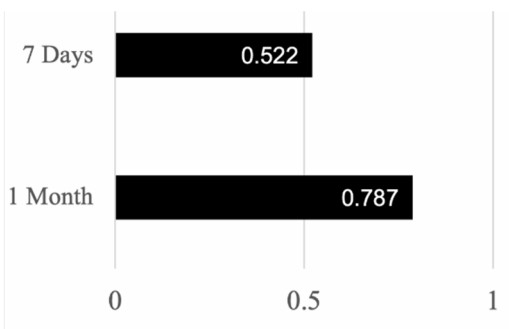

**Figure 17.** Correlation r with annual turnover.

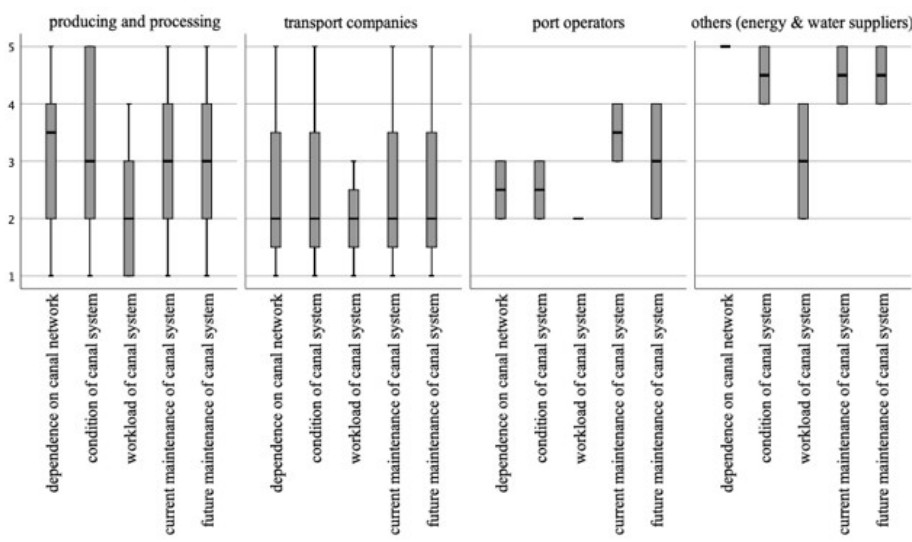

**Figure 18.** Risk perception depending on type of company.

**Step 3: Expert interviews**

To gather more detailed background information, we conducted two interviews in accordance with the hypotheses and the online questionnaire. Participants were asked about the following topics: the company's dependence on the waterway, the company's perception of risk, and the measures required in relation to the IWT.

The questions examined at what point in time closures threaten to disrupt a company's business operations severely and whether this impairment can be avoided or significantly reduced by a sufficiently long warning period. The responses provide an understanding of the background of the survey data. Moreover, we could also gain deeper insights into a central aspect of H2: maintenance today is inadequate with a backlog of modernization and the situation is expected to worsen due to increasing aging. However, the answers also indicate hope reflected by planned measures as shown in Figure 18.

**Step 4: Combined results**

We now turn to the relationship between IWT preference and criticality, which we find to be positively correlated. Our results are significant at an early warning time of 7 days (r = 0.682) and of 1 month (r = 0.743). The IWT preferences are derived from the proximity analysis and show high preference levels and thus the correlated high criticality for the sectors of mining and quarrying, coking plant and mineral oil processing, production of chemical and pharmaceutical products, metal production and processing, and energy supply.

To answer our research question *(2) What business decisions may result from lasting availability reductions of IWT?*, we proceed with a combined evaluation and obtained the following findings:

- Early warning time is of high importance for firms to be able to react to restrictions of IWT,
- Infrastructure disruptions hit firms especially hard due to a lack of road and rail capacities, and
- Whether the adverse effects of reduced infrastructure availability can be considerably reduced by sufficient early warning time depends on the vulnerability of the company.

## 5. Summary, Discussion, and Conclusions

In this contribution, we carried out a risk assessment of the waterway infrastructure as a barely studied transport system. We analyzed risk exposure stemming from infrastructure failure and had a focal look at the economic effects on potentially affected industries. From a methodological point of view, we used GIS-Analyses, economic statistics, and industry surveys which we applied to the case of NRW.

Analytical insights include substantial qualitative and quantitative impacts on industry and population that arise if waterways are not maintained. In detail, our elaborations on the economic risk potential show the significance of vulnerability and criticality of industrial sectors, and a predominance of location preferences for several industrial sectors, as Section 4.2.1 demonstrates. We elaborate that specific sectors are reliant on the functioning of IWT and thus on both accessibility and availability of this mode of transport. It becomes evident that non-existent or not usable capacities of alternative modes of transport pose a threat to business locations if the condition of waterways deteriorates significantly.

The proximity analysis, therefore, provides an overview of affected facilities in case of disruptive events. As an innovative approach, the analysis shows for the first time how business locations of different sectors are distributed in relation to existing access points of transportation modes. Moreover, the provided empirical studies (Section 4.2.2) contribute to the understanding of company decisions as a reaction to actual and perceived risk arising from the dependence on IWT. It is shown that modal shifts are a measure that is already taken by a vast majority, while the potential is highly restricted. This is primarily due to the capacities and availabilities of other transport modes. In combination with the consideration of IWT being hard to shift to the road as the highest $CO_2$ emitter among the modes of transport, and the long-term planning periods to achieve capacity increases on the railways, we conclude with the importance of ensuring a functioning infrastructure.

Accordingly, these results are of high interest to stakeholders from the industry, since they are enabled to assess the resilience of their business locations and SCs when considering our survey. Thus, those responsible for both infrastructure and industry should work together to enhance and preserve business locations. To apply effective holistic risk-avoiding and risk-preventing measures in the long-term, further research should also consider other external incentives for location relocation mechanisms and their interaction with the observed factor of infrastructure availability. Our research so far also does not provide one key figure in risk assessment since this is largely determined by the interpretation of the results, which would go beyond our research objective of understanding company decisions as reactions toward infrastructure failure.

The incorporation of empirical data and expert knowledge into the development of a risk-based maintenance strategy should be pursued, while an extension of the economic risk potential analysis of the provided survey toward further companies can provide additional insights. This could allow a more specific analysis and may refine possible gradations of risk and maintenance prioritization. Nevertheless, given the usual difficulties of data collection on a regional and local basis, we were able to extract and use a comprehensive and meaningful basis of regional and specific data to employ for our analyses.

**Author Contributions:** Conceptualization, R.W. and M.W.; methodology, F.N., R.W.; software, F.N.; validation, R.W. and M.W and F.S.; formal analysis, F.N., R.W.; investigation, F.N., R.W.; resources, F.S. and M.W.; data curation, F.N.; writing—original draft preparation, F.N. and R.W.; writing—review and editing, M.W., F.S. and R.W.; visualization, F.N. and R.W.; supervision, M.W. and F.S.; project

administration, R.W.; funding acquisition, F.S. and M.W. All authors have read and agreed to the published version of the manuscript.

**Funding:** This research was funded by BMBF, grant number 13N14697. The APC was funded by the KIT-Publication Fund of the Karlsruhe Institute of Technology.

**Institutional Review Board Statement:** Not applicable.

**Informed Consent Statement:** Informed consent was obtained from all subjects involved in the study.

**Data Availability Statement:** All data, which support the results of this study, are inidicated.

**Acknowledgments:** The authors want to thank the partners in the joint research project PREVIEW. The project is funded by the security research program (www.sifo.de accessed on 1 September 2022)) of the German Federal Ministry of Education and Research (BMBF). Special thanks go to the stakeholders and participants of the survey and interviews, who shared their expertise, thereby supporting this research. We acknowledge support by the KIT-Publication Fund of the Karlsruhe Institute of Technology.

**Conflicts of Interest:** The authors declare no conflict of interest.

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
