# Peer review of "Economic Risk Potential of Infrastructure Failure Considering In-Land Waterways"

_water, doi:10.3390/w14182874_

Round 1

Reviewer 1 Report

Authors in their paper perform a research on unreliability of transport infrastructures that can cause negative externalities for industries.  Their research is of good quality, paper is well written and well-structured.

However, I have several suggestions to improve the paper mostly focused on the interpretation of the authors arguments (for example, authors may used more own citations for improvement of their research). Also, another changes are necessary before the paper reaches the journal's standards and is accepted for publication.

There are several errors in formating, for example:

- line 40 and 41 - larger font used than in text;

- figures are once centred, then they are aligned to the left margin;

- consider number of pages. 

Author Response

Dear reviewer,

thank you very much for taking the time to assess our manuscript. We addressed your concerns as listed below and think, that our manuscript benefited from your comments

Reviewer Comment

Changes in manuscript

1.       Moderate English changes are required

We checked the language and adjusted several formulations

2.       Improvements on the interpretations of the authors arguments

We adjusted the section “summary, discussion and conlusion”

3.       line 40 and 41 - larger font used than in text;

We adjusted the font sizes

4.       figures are once centred, then they are aligned to the left margin

We adjusted the position of figures that were aligned to the left margin

5.       consider number of pages

Number of pages are integrated in the template

We would like to thank the referee again for taking the time to review our manuscript and remain with kind regards!

Reviewer 2 Report

The authors investigated economic risk potential of infrastructure failure while considering in land waterways. The paper suffers from technical contents of economic risk and reliability conceptions. Therefore, the paper needs major revisions before it is processed:

1-Lines 23-43: Literature review is rather poor. This section should be improved. Many recently-published papers can improve the quality of literature in terms of reliability of infrastructures:

-Pipe Break Rate Assessment While Considering Physical and Operational Factors: A Methodology Based on Global Positioning System and Data Driven Techniques

-A reliability-based probabilistic evaluation of the wave-induced scour depth around marine structure piles

2-The following statements should not be presented in such way:

We state the key questions of our research:  (1) Which industries and company-locations are directly affected by IWT failure? (2) What business decisions may result from lasting availability reductions of IWT?

3-Motivation and novelty should be described.

4-Which measures were applied to compute economical risk? This issue needs robust clarifications.

5-What is risk formulation for this research?? 

6-Line 244: "Case" is not proper title.

7-The proper title is "Economic risk potential of infrastructure failure while considering in land waterways".

Author Response

Dear reviewer,

thank you very much for taking the time to assess our manuscript. We addressed your concerns as listed below and think, that our manuscript benefited from your comments

Reviewer Comment

Changes in manuscript

1.       English language and style are fine/minor spell check required

We checked the language and adjusted several formulations

2.       Lines 23-43: Literature review is rather poor. This section should be improved. Many recently-published papers can improve the quality of literature in terms of reliability of infrastructures.

We inserted a brief section at the beginning of the Literature review. Nevertheless, we want to focus on the economic externalities and cascading effects, which are analyzed rather scarce. We hope this is getting clear from our formulations.

3.       The following statements should not be presented in such way: We state the key questions of our research:  (1) Which industries and company-locations are directly affected by IWT failure? (2) What business decisions may result from lasting availability reductions of IWT?

We think this poses the basis for the presented structured work and supports the traceability of the paper; Nevertheless, we are open to other proposals, if you have a specific idea.

4.       Motivation and novelty should be described.

We specified motivation and novelty within the Introduction

5.       Which measures were applied to compute economical risk? This issue needs robust clarifications.

At several points, we inserted a remark that we focus on cause-effect chains rather than measures as posing one key figure. Hence, we provide several key figures with the potential to anticipate future reactions of economic decision-makers, but we do not pre-empt a holistic risk quantification, since this must be set in relation to further dimensions of criticality and vulnerability, which is out of scope from our research.

6.       What is risk formulation for this research??

We highlighted the query within section 2.2.2; ~Line 101

7.       Line 244: "Case" is not proper title.

We adjusted the title of the corresponding section

8.       The proper title is "Economic risk potential of infrastructure failure while considering inland waterways".

We did not adjust the title since we think that the fill word “while” is not needed

We would like to thank the referee again for taking the time to review our manuscript and remain with kind regards!

Reviewer 3 Report

Plagiarism was detected, which is within the acceptable limit.

Author Response

Dear reviewer,

thank you very much for taking the time to assess our manuscript. We addressed your concerns as listed below and think, that our manuscript benefited from your comments

Reviewer comment

Changes in manuscript

1.       English language and style are fine/minor spell check required

We checked the language and adjusted several formulations

2.       Plagiarism was detected, which is within the acceptable limit.

We did make minor corrections here

We would like to thank the referee again for taking the time to review our manuscript and remain with kind regards

Round 2

Reviewer 2 Report

The introduction section and literature review is rather superficial

Author Response

Dear reviewer,

thank you very much for taking the time to assess our manuscript. We addressed your concerns as stated below and think, that our manuscript benefited from your comments.

We extended the literature review to include more current and relevant sources and linked to them in order to delimit our research. We precised our research design and expanded the introduction section. Here, we refer to the literature review in order to not duplicate too many insights. We hope, that we could realize conceivable improvements in our research but are still open for any further suggestions.

We would like to thank the referee again for taking the time to review our manuscript and remain with kind regards